The impact of multidomain interventions on cognitive and physical function in older adults with subjective cognitive decline: a meta‑analysis and systematic review

Yi Qing 1
Wang Wei 2
http://orcid.org/0000-0002-8623-6692 Qi Yufei 3
Yang Chengwei 1
Sui Mengyun 4
Meng Kun 5 mengkun1108@163.com
Zhao Shanguang 6 sgzhao@shmtu.edu.cn
1 Faculty of Sports and Exercise Science, Universiti Malaya , Kuala Lumpur , Malaysia
2 Center for Health Policy Research and Evaluation, School of Public Administration and Policy, Renmin University , Beijing , China
3 Department of Physical Education and Research, Central South University , Changsha, Hunan , China
4 Shanghai Municipal Center for Disease Control & Prevention , Shanghai , China
5 School of Physical Education, Hunan First Normal University , Changsha, Hunan , China
6 Department of Physical Education, Shanghai Maritime University , Shanghai , China
Hepsomali Piril
Electronic publication date: 2025 Jun 26
Publication date: 2025
Volume: 13
Electronic Location ID: e19588
Received 2024 Sep 11; Accepted 2025 May 20
Copyright: © 2025 Yi et al.
Copyright year: 2025
Copyright holder: Yi et al.
License: This is an open access article distributed under the terms of the Creative Commons Attribution License, which permits unrestricted use, distribution, reproduction and adaptation in any medium and for any purpose provided that it is properly attributed. For attribution, the original author(s), title, publication source (PeerJ) and either DOI or URL of the article must be cited.
License URL: https://creativecommons.org/licenses/by/4.0/

Keywords: Multicomponent intervention, Nonpharmacological intervention, Cognitive performance, Cognitive decline, Healthy aging

Funding: The authors received no funding for this work.

==============================
Background

This study aimed to examine the efficacy of multidomain interventions compared to control and nutritional interventions in older adults with subjective cognitive decline (SCD). Four databases were searched for relevant literature.

Methodology

PubMed, Embase, Cochrane Library, and Web of Science databases were searched for relevant studies. Randomized controlled trials that examined the effects of multidomain interventions on cognitive and physical function in older adults with SCD were included.

Results

This study included six eligible studies with 1,767 participants. The results indicated that multidomain interventions significantly improved executive function and memory but did not significantly impact global cognition and physical performance compared to the control group. In addition, significant enhancements were observed in executive function and memory when compared to single nutritional interventions.

Conclusions

The findings revealed that multidomain interventions could effectively improve executive function and memory in older adults with SCD. Further studies with robust designs, particularly those comparing single-domain interventions, are needed to investigate the effects and underlying mechanisms.

Introduction

With the global population aging, age-related cognitive decline has become a major public health concern. It is predicted that 152.8 million individuals will experience dementia by 2050 (GBD 2019 Dementia Forecasting Collaborators, 2022), leading to a considerable impact on the socio-economic system (Alzheimer’s Association, 2023). Subjective cognitive decline (SCD) is defined as self-reported cognitive complaints in the absence of objectively measurable cognitive deficits on neuropsychological assessments (Jessen et al., 2014). SCD is commonly regarded as the earliest precursor stage of Alzheimer’s disease, and as an intermediate stage between normal aging and mild cognitive impairment (MCI). A meta-analysis of 28 cohort studies found that individuals with SCD had a two-fold relative risk of 2.07 (95% CI [1.77–2.44]) for dementia conversion compared to those with healthy cognitive aging. The annual conversion rates were 2.33% for SCD and 6.67% for MCI (Mitchell et al., 2014). Those findings highlight the importance of early interventions, with SCD offering a promising window to delay further cognitive decline.

Currently, the efficacy of pharmacological interventions for cognitive decline remains unproven, and they commonly have side effects (Livingston et al., 2020). In contrast, non-pharmacological interventions have demonstrated promising effects without adverse effects for older adults with SCD (Smart et al., 2017). Among these, nutritional, exercise, and multidomain interventions are three effective non-pharmacological approaches to improve cognitive function. Nutrition is a key modifiable factor in preserving brain health during aging (Flanagan et al., 2020; Melzer et al., 2021; Morris, 2012). The cognitive benefits of nutritional interventions have been examined through various nutrients and dietary patterns, including folic acid (Gil Martínez, Avedillo Salas & Santander Ballestín, 2022), vitamin D (Jiang et al., 2023), and omega-3-polyunsaturated fatty acid (Cherbuin, Anstey & Baune, 2017), the Mediterranean diet (Valls-Pedret et al., 2015), the Dietary Approaches to Stop Hypertension diet (DASH) (Smith et al., 2010), and ketogenic diet (Krikorian et al., 2012). Nutritional interventions improve cognitive function primarily through the supplementation of specific nutrients. Diets rich in antioxidants, healthy fats, and anti-inflammatory compounds—including the Mediterranean and DASH diets—have been shown to reduce oxidative stress and inflammation, improve vascular health, and protect against age-related neurodegeneration (Morris et al., 2015; Scarmeas et al., 2007).

Physical exercise is another well-established modifiable factor in preserving cognitive health. Aerobic (Kamijo et al., 2009; Young et al., 2015), resistance (Coelho-Junior et al., 2022; Liu-Ambrose et al., 2010), multicomponent exercises (Li et al., 2021), and mind-body exercises (Wu et al., 2019; Xia et al., 2019) are common strategies in preserving cognitive function and preventing cognitive decline in older adults. Physical activity may indirectly influence cognitive function by improving overall health and enhancing neuroplasticity (Bherer, 2015; Bruderer-Hofstetter et al., 2018). Similarly, cognitive interventions, based on the theories of neuroplasticity, have proven to be effective strategies for mitigating cognitive decline (Liu et al., 2021a; Marlats et al., 2020). Some systematic reviews have demonstrated the positive effects of multidomain interventions in enhancing cognitive outcomes (Gavelin et al., 2020; Sood et al., 2019). Cognitive interventions provide a targeted approach to improve brain function by directly stimulating and challenging cognitive processes. Cognitive interventions may directly impact cognitive function by enhancing cognitive plasticity, cognitive reserve, and neural connectivity (Cheng, 2016; Li et al., 2017; van Balkom et al., 2020).

Multimodal intervention strategies have gained increasing attention for their potential synergistic effects. Multidomain interventions are defined as interventions that incorporate at least two different domains, such as nutritional, exercise, cognitive, or psychosocial interventions (Bevilacqua et al., 2022). Currently, the more effective multidomain intervention patterns are those that include nutritional, exercise, and cognitive interventions (Castro et al., 2023), as demonstrated by large-scale trials such as the Finnish Geriatric Intervention Study to Prevent Cognitive Impairment and Disability (FINGER) (Ngandu et al., 2015) and Multidomain Alzheimer Preventive Trial (Andrieu et al., 2017). The Japan-Multimodal Intervention Trial for Prevention of Dementia (J-MINT) is also a part of the global FINGERS network. The study found that, after 18 months, the intervention group showed a significant improvement in cognitive composite scores (Oki et al., 2024). In addition, MIND-AD trials targeting the early stages of Alzheimer’s patients demonstrated that interventions combining lifestyle modification and drug supplementation reduced the rate of cognitive decline after 6 months (Thunborg et al., 2024). Furthermore, a meta-analysis suggested that multidomain interventions combined with physical exercise might benefit overall cognitive function in older adults (Reparaz-Escudero et al., 2024). Conversely, the AgeWell.de trial adjusted the FINGER intervention model to test a multimodal intervention (including optimized nutrition, medication, and physical, social, and cognitive activity), but no effect on global cognitive performance was detected (Zülke et al., 2024). Similarly, another study found no evidence supporting the effectiveness of multidomain interventions for preventing dementia (Hafdi, Hoevenaar-Blom & Richard, 2021). Overall, the effects of multidomain interventions on cognitive function in older adults remain controversial (Hafdi, Hoevenaar-Blom & Richard, 2021; Oki et al., 2024; Thunborg et al., 2024; Zülke et al., 2024), and comparisons with single-domain interventions are still limited (Dedeyne et al., 2017; Fessel et al., 2017).

Although some meta-analyses and reviews have reported positive effects of multidomain interventions on cognitive and physical functions (Ahn et al., 2022; García-Llorente et al., 2024; Liao, Shen & Li, 2023; Mohanty & Kumar, 2022), the varied components of these included interventions may introduce significant heterogeneity. In addition, these studies included designs other than randomized controlled trials (RCTs) (Fessel et al., 2017; Mohanty & Kumar, 2022) and lacked rigorous inclusion criteria for study populations (Liu et al., 2021b; Noach et al., 2023). To our knowledge, only one review has examined the effects of multidomain interventions on individuals with SCD, targeting those over 45 years old and including RCTs, cohort studies, and reviews (Mohanty & Kumar, 2022). However, no meta-analysis with robust included criteria exists for older adults with SCD. Therefore, this study defines multidomain interventions as a combination of nutrition, physical exercise, and/or cognitive training, and includes only RCTs to examine their effects on cognitive and physical function in older adults with SCD.

This is the first meta-analysis with a robust design to examine the effectiveness of the multidomain interventions on cognitive and physical functions among older adults with SCD. This study aimed to evaluate the effects of multidomain interventions compared to the control and nutritional interventions.

Materials and Methods

This meta-analysis complied with the Preferred Reporting Items for Systematic Reviews and Meta Analyses (PRISMA) guidelines (Liberati et al., 2009) and PRISMA extension for meta-analysis (Hutton et al., 2015). The present study has been registered on the International Platform of Registered Systematic Review and Meta-analysis Protocols (INPLASY, registration number: INPLASY202460020).

Search and selection strategies

Relevant studies were identified through searches in PubMed, Embase, Cochrane Library, and Web of Science databases on March 6, 2024. To ensure the inclusion of the most recent research, we conducted an updated search on December 31, 2024. There were no restrictions on publication language, date, or type. To enhance the comprehensiveness of the search terms, we referred to prior studies (Liu et al., 2021b; Roberts et al., 2022). The final search strategies employed Boolean operators “OR” and “AND” in conjunction with terms such as “multidomain interventions”, “nutritional intervention”, “exercise intervention”, “SCD”, and “aging”. Detailed search strategies are provided in the Supplemental Material S1. Additionally, we manually reviewed the references of included publications to ensure a comprehensive search.

EndNote 20 was used to manage, categorize, and deduplicate the records. Initially, two researchers (Q.Y. and Y.Q.) independently screened the titles and abstracts of the documents. Subsequently, the full texts of the selected papers were carefully reviewed for eligibility. Discrepancies were reviewed by a third researcher (C.Y.) until a consensus was reached.

Selection criteria

Types of studies

Published, peer-reviewed documents of RCTs and cluster-RCTs that examined the effects of multidomain interventions on cognitive and physical functions in older adults with SCD were included. Excluded studies encompassed unpublished articles, reviews, study protocols, case reports, case studies, theses, dissertations, and book chapters.

Types of participants

Eligible studies focus on older adults with SCD aged over 55. The participants should have self-reported cognitive decline but perform normally on standardized cognitive tests and do not meet diagnostic criteria for MCI or dementia.

Types of interventions

Intervention protocols should include exercise and nutritional interventions with or without multidomain interventions. Given the wide variety of multidomain interventions, including various components would increase the heterogeneity of the pooled results, making it challenging to examine the effects of a specific type of multidomain intervention. Therefore, this study focused on a more effective multidomain intervention pattern, incorporating exercise, nutrition, and multidomain interventions (Castro et al., 2023). Notably, due to the limited number of studies including all three intervention components, the intervention programs in this study included exercise and nutrition, with or without cognitive interventions.

Types of comparisons

The comparisons should meet one of the following criteria: (1) a control group (e.g., placebo, maintain usual life habits or health education); (2) a single domain intervention (either nutritional interventions, exercise intervention, or multidomain interventions).

Types of outcomes

Studies had to include at least one cognitive or physical outcome. Primary outcomes are cognitive function, such as global cognition or specific cognitive domains (e.g., executive function, memory). Secondary outcomes are physical function, such as strength, endurance, and mobility. Furthermore, the outcomes should assess changes from baseline to post-intervention.

Data extraction

Data was independently extracted by the two researchers using a self-developed standardized form. Any disagreements were resolved by the third researcher. Information such as the author’s name, publication year, country, sample size, age, types of intervention, length, frequency, duration of intervention, types of control, and outcomes was collected from the included studies.

Quality assessment

Quality assessment was independently conducted by the two researchers using Review Manager 5.4. Any disputes were confirmed by the third researcher. The qualitative assessment focused on seven aspects of the RCTs: random sequence generation (selection bias), blinding of participants and personnel (performance bias), blinding of outcome assessment (detection bias), allocation concealment (selection bias), incomplete outcome data (attrition bias), selective reporting (reporting bias), and other biases. Each index was rated as “high risk,” “low risk,” or “unclear risk.”

Statistical analysis

Data analysis was conducted using Review Manager 5.4 and Stata 18. The analysis included calculating the combined effect size, assessing heterogeneity, and generating forest plots. Cognitive function and physical function were treated as continuous variables. The pooled effects of continuous variables were assessed using either mean difference (MD) or standardized mean difference (SMD) with 95% confidence intervals (95% CI). MD was used for variables measured by the same method, and SMD for those measured by different methods. According to Cohen’s guidelines for effect sizes, 0.20, 0.50, and 0.80 represent the thresholds for small, medium, and large effect sizes, respectively (Cohen, 2013). Due to the inherent heterogeneity stemming from the subjectivity of SCD and the variability across intervention methods, a random-effects model was employed for all analyses (Borenstein et al., 2021). Heterogeneity was assessed using the I2 statistic, with levels rated as low (25%), moderate (50%), and high (75%). Sensitivity analysis was performed when I2 > 50% (Coryn, 2011). Additionally, this study used Stata to perform the Begg test to identify publication bias (Cohen, 2016).

Results

Study selection

As shown in Fig. 1, the initial search identified 5,402 records. After removing 1,080 duplicates, the two researchers screened the remaining 4,322 documents by reading the titles and abstracts. A total of 133 articles were then selected for full-text review, resulting in the exclusion of 128 articles that did not meet the inclusion criteria. The specific reasons for exclusion are provided in Fig. 1. Ultimately, five studies met the inclusion criteria. In addition, one eligible article was identified by searching the references of the included studies. Thus, a total of six articles were included in the study.

Figure 1 PRISMA flow diagram.

Study characteristics

Table 1 presents six eligible studies involving 1,767 participants with a higher proportion of female participants than male participants. Regarding the nutritional interventions, one study (Andrieu et al., 2017) requested participants to take daily omega-3 supplements, with 800 mg of docosahexaenoic acid (DHA) and up to 225 mg of eicosapentaenoic acid (EPA) per day with dietary guidance. One study (Macpherson et al., 2022) administered the intervention through a daily drink supplement (5 g of omega-3 powder, 1,000 IU vitamin D3, and 25 g whey protein concentrate (WPC) 80%). Additionally, one study (Chatterjee et al., 2022) utilized the Mediterranean diet, and another study (Blumenthal et al., 2020) employed the DASH diet. Moreover, two studies (Barreto et al., 2021; Liu et al., 2023) offered nutritional counseling to assist participants in maintaining healthy dietary habits.

Table 1 Basic information of eligible literature.

Study:(Author; year; country)	Participants Characteristics	Multidomain intervention design	Comparisons	Outcomes/Findings	
Size	Mean age	Female (%)	Exercise intervention (length; frequency)	Nutritional intervention	Cognitive intervention	Duration	
Barreto et al. (2021); France	109	MI: 75.2 ± 5.7; CG: 73.2 ± 5.3	MI: 51.7%; CG: 63.3%	Type: AE;	Nutritional advice every 15 days; and nutritional planning for people at risk of nutritional deficiencies.	2 times/week computerized cognitive training using Neuropeak: main contents are executive function training.	24 weeks	CG: Placebo	MI group significantly improved in memory function and SPPB, but no significant improvement in global cognition.	
RE (three types); BE (three types)	
Frequency: 2 times/week	
Intensity: the number of both sets and repetitions per set varied according to individual’s physical function	
Time: NM	
Duration: 24 weeks	
Blumenthal et al. (2020); USA	160	Overall: 65.4 ± 6.8	Overall: 66%	Type: AE (walking or stationary biking)	DASH diet and diet counseling (first 3 months: weekly section, remaining 3 months: bi-weekly section).	NC	24 weeks	EI: AE; NI: DASH diet and counseling; CG: Maintaining dietary and exercise habits.	Compared to the NI group, MI group showed significant improvement in executive function and memory.	
Frequency: 3 times/week	
Intensity: Moderate intensity	
Time: 35 min	
Duration: 24 months	
Andrieu et al. (2017); France	1,135	MI: 75.4 ± 4.4; NI: 75.6 ± 4.7; CG: 75.1 ± 4.3	MI: 61%; NI: 64%; CG: 66%	Type: AE (Walking)	Two capsules: Omega 3—PUFA (consumed daily; 800 mg DHA and up to 225 mg EPA per day); and nutritional advice based on dietary guidelines of French.	First 2 months (first month (twice a week), second month (once a week)): 60 min of reasoning and memory training per session. Remaining 34 months: a monthly session to reinforce information.	144 weeks	NI: Omega 3-PUFAs; CG: Placebo	Compared to CG, MI and NI, did not significantly improved memory and SPPB.	
Frequency: First 2 months: 5 days a week; Remaining 34 months: One monthly session to reinforce information.	
Intensity: Moderate intensity	
Time: First 2 months: 30 min. Remaining 34 months: Duration of monthly sessions not specified, but focused on reinforcing information.	
Duration: 3 years	
Chatterjee et al. (2022); India	30	MI: 65.2 ± 3.73; CG: 71.1 ± 6.93	MI: 33.3%; CG: 40%	Type: AE (walking or stationary biking) and RE (extremity and core strengthening)	Mediterranean equivalent diet (Modified Mediterranean diet pattern from FINGER).	Supervised CBCT 2 times/week, a total of 40 sessions: RehaCom software with 29 tasks was used for cognitive training.	24 weeks	CG: Awareness instructions for brain-stimulating activities	MI significantly improve executive
function, but did not on memory.	
Frequency: 2 times/week	
Intensity: AE: 40–80% HR Max; RE: 40–80% Repetition Maximum	
Time: 60 min	
Duration: 24 months	
Liu et al. (2021b); China	192	MI: 69.7 ± 6.0; CG: 73.1 ± 5.7	MI: 74.4%; CG: 66%	Type: AE, RE and BE and homework	Nutrition counseling: Maintaining healthy dietary habits (one education session and two individual visits tailored to each participant).	Supervised cognitive training consisted of 20 group sessions and daily homework.	36 weeks	CG: Usual care without intervention components	MI group showed a significant improvement in global cognition immediately after the intervention, but there was no significant improvement in executive function and memory.	
Frequency: AE, RE and BE (2 times/week); homework (everyday)	
Intensity: NM	
Time: NM	
Duration: 15 weeks	
Macpherson et al. (2022); Australia	147	MI: 70 ± 6.3; CG: 70.5 ± 5.9	MI: 70%; CG: 70%	Type: AE (treadmill, bike, stepper, or rower) and RE (6 resistance and 2 core exercises)	Supplement drink: Consumed daily including 5 g of omega-3 powder (900 mg EPA and 600 mg DHA per day), 1,000 IU vitamin D3, and 25 g of WPC 80%.	NC	24 weeks	CG: Placebo	MI group showed significant within group improvement in executive function, with no change in global cognition.	
Frequency: 2 times/week	
Intensity: AE: rate of perceived exertion (RPE) of 5–8 (hard to very hard); RE: weekly progressive overload (increments of 2–10%)	
Time: 60 min	
Duration: 24 weeks	
Note:

MI: multidomain interventions; NI: nutritional interventions; EI: exercise intervention; CG: control group; CF: cognitive function; PF: physical function; AE: aerobic exercise; RE; resistance exercise; BE: balance exercise; NC; not clear; DHA: docosahexaenoic acid; EPA: eicosapentaenoic acid; PUFA: polyunsaturated fatty acid; WPC: whey protein concentrate; FINGER: Finnish geriatric intervention study to prevent cognitive impairment and disability; CBCT: computer-based cognitive therapy.

For the exercise intervention, the interventions took place one to five times per week, with session lengths ranging from 30 to 60 min, and the durations varied from 24 weeks to 3 years. Specifically, two studies (Barreto et al., 2021; Liu et al., 2023) involved aerobic, resistance, and balance exercise, whereas another two studies (Chatterjee et al., 2022; Macpherson et al., 2022) incorporated both aerobic and resistance exercises. Regarding cognitive interventions, two studies (Barreto et al., 2021; Chatterjee et al., 2022) employed computerized cognitive training, focusing mainly on memory, executive functions, and reasoning. In addition, two studies (Blumenthal et al., 2020; Macpherson et al., 2022) did not include cognitive interventions.

Regarding the comparisons, three studies (Andrieu et al., 2017; Barreto et al., 2021; Macpherson et al., 2022) used a placebo control group, while two studies (Andrieu et al., 2017; Blumenthal et al., 2020) included a nutrition group. Two studies (Blumenthal et al., 2020; Liu et al., 2023) included a control group that maintained daily living habits, and one study (Blumenthal et al., 2020) included an exercise intervention group. In terms of outcomes, four (Andrieu et al., 2017; Barreto et al., 2021) studies included both cognitive and physical function outcomes, while the remaining two studies (Blumenthal et al., 2020; Chatterjee et al., 2022; Liu et al., 2023; Macpherson et al., 2022) focused only on cognitive function outcomes.

Risk of bias of included studies

As depicted in Fig. 2, the bias assessment follows the guidelines outlined by the Cochrane Collaboration (Higgins & Green, 2008). The graph utilizes green, yellow, and red colors to represent low risk, unclear risk, and high risk, respectively. Among the six included studies, five studies (Andrieu et al., 2017; Barreto et al., 2021; Chatterjee et al., 2022; Liu et al., 2023; Macpherson et al., 2022) reported detailed methods for generating random sequences, and three (Andrieu et al., 2017; Barreto et al., 2021; Chatterjee et al., 2022) provided the methods for allocation concealment. The blinding of participants and personnel in the four studies was identified as low risk (Andrieu et al., 2017; Barreto et al., 2021; Chatterjee et al., 2022; Macpherson et al., 2022). Five studies (Andrieu et al., 2017; Barreto et al., 2021; Blumenthal et al., 2020; Liu et al., 2023; Macpherson et al., 2022) described the use of blinding of outcome assessment. Notably, three studies (Barreto et al., 2021; Liu et al., 2023; Macpherson et al., 2022) were at high risk of selective reporting. Lastly, all six studies (Andrieu et al., 2017; Barreto et al., 2021; Blumenthal et al., 2020; Chatterjee et al., 2022; Liu et al., 2023; Macpherson et al., 2022) reported no risk of incomplete outcome data and other bias.

Figure 2 Risk of bias evaluation (Andrieu et al., 2017; Barreto et al., 2021; Blumenthal et al., 2020; Chatterjee et al., 2022; Liu et al., 2021b; Macpherson et al., 2022).

Effects of multidomain interventions vs the control group

Global cognition

Four studies (Andrieu et al., 2017; Barreto et al., 2021; Liu et al., 2023; Macpherson et al., 2022) investigated the effects of multidomain interventions on global cognition. Global cognition was assessed using the Mini-Mental State Examination (MMSE) or Montreal Cognitive Assessment (MoCA). This analysis used the SMD due to the different assessment tools used across studies. The pooled results indicated that multidomain interventions did not significantly improve global cognition with a small effect size (SMD = 0.08, 95% CI [−0.03 to 0.20], p = 0.15, Fig. 3).

Figure 3 Forest plot of multidomain interventions vs control group on global cognition (Andrieu et al., 2017; Barreto et al., 2021; Liu et al., 2021b; Macpherson et al., 2022).

Executive function

Three studies (Chatterjee et al., 2022; Liu et al., 2023; Macpherson et al., 2022) evaluated the effectiveness of multidomain interventions vs the control group. As shown in Fig. 4, multidomain interventions significantly improved executive function, with a moderate effect size (SMD = −0.24, 95% CI [−0.45 to −0.03], p = 0.02).

Figure 4 Forest plot of multidomain interventions vs control group on executive function (Chatterjee et al., 2022; Liu et al., 2021b; Macpherson et al., 2022).

Memory

Figure 5 demonstrates four studies (Andrieu et al., 2017; Barreto et al., 2021; Chatterjee et al., 2022; Liu et al., 2023) that examined the memory gains of multidomain interventions. The pooled analysis indicated that multidomain interventions greatly enhanced memory with a moderate effect size (SMD = 0.45, 95% CI [0.03–0.88], p = 0.04).

Figure 5 Forest plot of multidomain interventions vs control group on memory (Andrieu et al., 2017; Barreto et al., 2021; Blumenthal et al., 2020; Chatterjee et al., 2022; Liu et al., 2021b).

Physical function

Two studies (Andrieu et al., 2017; Barreto et al., 2021) assessed the efficacy of the multidomain interventions on physical function using the Short Physical Performance Battery. Pooled results were not statistically significant with a small effect size (MD = 0.02, 95% CI [−0.25 to 0.29], p = 0.88, Fig. 6).

Figure 6 Forest plot of multidomain interventions vs control group on physical function (Andrieu et al., 2017; Barreto et al., 2021).

Effects of multidomain interventions vs nutritional interventions

For comparisons with single domain intervention, due to the limited number of studies, the meta-analysis could not be performed on separate exercise interventions and cognitive interventions. Thus, we only compared the effects of multidomain interventions with those of single nutritional interventions.

Executive function

Two studies (Andrieu et al., 2017; Blumenthal et al., 2020) evaluated the effectiveness of multidomain interventions vs nutritional interventions. As shown in Fig. 7, multidomain interventions produced a significantly enhanced executive function, with a moderate effect size (SMD = −0.31, 95% CI [−0.59 to −0.04], p = 0.03).

Figure 7 Forest plot of multidomain interventions vs control group on executive function (Andrieu et al., 2017; Blumenthal et al., 2020).

Memory

Figure 8 shows that two studies (Andrieu et al., 2017; Blumenthal et al., 2020) examined the effects of multidomain interventions on memory. The pooled results reported that multidomain interventions significantly improved memory with a moderate effect size (MD = −4.07, 95% CI [−6.71 to −1.31], p = 0.004).

Figure 8 Forest plot of multidomain interventions vs nutritional interventions on memory (Andrieu et al., 2017; Blumenthal et al., 2020).

Sensitivity analysis and publication bias

Since the heterogeneity of multidomain interventions compared to the control group on memory was greater than 50% (I2 = 85%), sensitivity analyses were conducted to assess the impact of each study on heterogeneity. As shown in Table 2, I2 value and p of I2 changed significantly when the article of Chatterjee et al. (2022) was excluded (Supplemental Material S2). Since only two studies compared combined and nutritional interventions on executive function, a sensitivity analysis was unnecessary. Due to most of the indicators included limited studies, this study only conducted publication bias tests on indicators with three or more included articles. Begg’s test (global cognition: p = 0.089, memory: p = 0.296, executive function: p = 0.089), all confirmed that there is no statistically significant publication bias (Supplemental Material S3–S5).

Table 2 Sensitivity analysis of executive function (Andrieu et al., 2017; Barreto et al., 2021; Chatterjee et al., 2022; Liu et al., 2021b).

Study omitted	SMD	95% CI	p	I2 (%)	p	
Andrieu 2017	0.74	[0.00–1.49]	0.0003	88	0.05	
Barreto 2021	0.55	[–0.04 to 1.13]	<0.0001	90	0.07	
Chatterjee 2022	0.13	[0.01–0.26]	0.36	3	0.04	
Liu 2022	0.69	[−0.03 to 1.41]	<0.0001	90	0.06	
Note:

SMD: Standardized mean difference; 95% CI: 95% confidence interval; I2 (%): I-squared, a measure of heterogeneity.

Discussion

To our knowledge, this is the first study to examine the efficacy of multidomain interventions on both the cognitive and physical functions of older adults with SCD. The results indicated that multidomain interventions significantly improved executive functions and memory, but did not impact global cognition and physical performance when compared to the control group. When compared to nutritional interventions, our study demonstrated that multidomain interventions significantly enhanced executive function and memory.

This study demonstrated that multidomain interventions have a moderate impact on executive function and memory compared to the control group, which may hold clinical relevance for older adults with SCD. These findings align with the review by Ahn et al. (2022), which included seventeen RCTs examining the effects of multidomain interventions on executive function and episodic memory in older adults without dementia (Ahn et al., 2022). Notably, executive function and memory have complex subcomponents (Wang & Dong, 2018), and different reviews may use various tools to assess these components, which would increase the heterogeneity of pooled results. Regarding global cognition, our study found that multidomain interventions could not significantly improve global cognition, contrary to previous results (Ahn et al., 2022; Liao, Shen & Li, 2023). A possible explanation is that the intervention period in this study was relatively short, which may have limited the manifestation of intervention effects. In addition, the limited number of included studies may also influence the pooled results.

In comparison to single-domain interventions, we only compared the effects of multidomain interventions with single nutritional interventions due to limited included studies. Our results indicated that multidomain interventions effectively enhanced executive function and memory with a moderate effect size, suggesting a possible synergistic moderate effect. A review reported that multidomain interventions are more promising than single-domain interventions, which supports our results (Fessel et al., 2017). However, our results should be interpreted with caution, as they are based on only two studies (Andrieu et al., 2017; Blumenthal et al., 2020). Another review also highlighted that the evidence for the beneficial effects of multidomain interventions is limited (Dedeyne et al., 2017). Future, more rigorously designed studies are needed to compare the effects of multidomain interventions with single-domain interventions, particularly emphasizing comparisons between exercise and cognitive interventions.

Mechanisms through which multidomain interventions exert beneficial effects on cognitive function remain unclear. Current research mainly explains the mechanisms through three aspects: (1) increasing cerebral blood flow, (2) modifying neurotransmitter release, and (3) modifying the structure and functional activities of the central nervous system (Gligoroska & Manchevska, 2012; Serra et al., 2020). Specifically, animal model studies have shown that increased physical exercise can promote hippocampal growth (van Praag et al., 2005), grey matter increase (Sumiyoshi et al., 2014), synaptogenesis (Jiang et al., 2024), and angiogenesis (Pereira et al., 2007), as well as induce the release of brain-derived neurotrophic factor (Almeida et al., 2015) and insulin-like growth factor-1 (Carro et al., 2000). In human studies, physical exercise has also been found to influence grey matter volume (Arenaza-Urquijo et al., 2017), cortical volume (Clark et al., 2019), and hippocampal volume (Ten Brinke et al., 2015), while also promoting the release of brain-derived neurotrophic factor (Marston et al., 2017). Nutritional interventions may help delay cognitive decline and the progression of dementia by improving cell membrane fluidity and vascular endothelial function (Howe et al., 2018; Teixeira et al., 2019), reducing inflammation and oxidative stress (Allès et al., 2012; Miquel et al., 2018; Tang et al., 2015), and promoting neurogenesis and neuronal connectivity (Dauncey, 2009; Gomez-Pinilla & Tyagi, 2013). Regarding cognitive interventions, Kempermann, Kuhn & Gage (1997) demonstrated in a mouse model that enriched environments protect white matter integrity by enhancing neurogenesis (Kempermann, Kuhn & Gage, 1997). Another study observed similar brain volume changes in older adults after cognitive training (Tait et al., 2017).

Another question worth investigating is why the cognitive effects of multidomain interventions are superior to those of single-domain interventions. One possible reason is that nutrition, exercise, and cognitive interventions have complementary and interactive effects. For example, regarding increased cerebral blood flow, consuming foods rich in antioxidants can improve endothelial function and reduce arterial stiffness (Howe et al., 2018), while regular participation in aerobic exercise can slow the decline in cerebral hemodynamics, such as cerebral arterial blood flow velocity and mean arterial pressure, associated with aging (Ainslie et al., 2008; Bailey et al., 2013). These two interventions synergistically improve cerebral blood flow, thereby enhancing nutrient delivery and promoting the release of neurotransmitters that aid cognition. Moreover, there are complementary mechanisms between exercise and cognitive interventions. For example, existing studies have shown that exercise supports the proliferation and division of neuronal cells (Hötting & Röder, 2013; Shors et al., 2012), while cognitive interventions help to promote the survival of these newborn cells (Kempermann, Kuhn & Gage, 1997).

Concerning physical function, the present study indicated that multidomain interventions could not significantly improve physical performance compared to the control group. This contrasts with findings from two review studies that indicated that multidomain interventions could improve physical performance compared to the control group (Giné-Garriga et al., 2015; Liao, Shen & Li, 2023). The possible explanations for our study, in which multidomain interventions did not significantly improve physical function, are that the intervention format in one study was 30 min of home-based exercise (Andrieu et al., 2017), the intervention frequency in the other study was twice a week (Barreto et al., 2021), and the intervention was not supervised. Such designs may not fully capture the benefits of exercise intervention. For other physical outcomes, some review studies demonstrated that multidomain interventions could effectively enhance strength, speed, and stability (Dedeyne et al., 2017; García-Llorente et al., 2024). For example, one review suggested that multidomain interventions were effective in improving muscle strength in (pre)frail elderly compared to single domain interventions (Dedeyne et al., 2017). However, another review found no significant difference in those outcomes when compared to exercise or nutritional interventions alone (Pérez-Bilbao et al., 2023). As a result, more research is needed to investigate the effects of multidomain interventions on physical function.

The American College of Sports Medicine (ACSM) and the American Heart Association (AHA) recommend that older adults engage in moderate-intensity aerobic exercise for 30 min five times per week or vigorous-intensity aerobic exercise for 20 min three times per week. In addition, older adults should perform moderate-intensity resistance training twice weekly. Moreover, ACSM and AHA suggest that incorporating balance and flexibility exercises can provide additional benefits (Nelson et al., 2007). Among the six studies included in this study, four implemented both aerobic and resistance training (Barreto et al., 2018; Chatterjee et al., 2022; Liu et al., 2023; Macpherson et al., 2022), while two focused solely on aerobic exercise (Andrieu et al., 2017; Blumenthal et al., 2020). All included studies met the recommended duration of physical activity, though only one study met the ACSM-recommended frequency of five sessions per week (Andrieu et al., 2017). Given the complexity of multidomain interventions, it is challenging to implement five exercise sessions per week due to the need to include multiple interventions. Notably, the expert consensus guidelines on exercise recommendations for older adults indicate that engaging in physical activity three times a week also shows significant benefits (Izquierdo et al., 2021). Therefore, future multidomain interventions should incorporate exercise sessions at a frequency of three or more times per week.

Strengths and limitations

This study has three main advantages. First, most studies included in the analysis are high-quality RCTs, ensuring the reliability of the intervention effects. Second, strict population inclusion criteria were applied, focusing exclusively on older adults with SCD. This distinguishes this study from most studies that include elderly individuals with varying cognitive function status, potentially introducing bias into the results. Third, this study assessed both cognitive and physical function indicators, providing a comprehensive evaluation of the effects of multidomain interventions.

This study also has three limitations. First, the included studies exhibit considerable variability in the characteristics of the multidomain interventions, such as different types of exercise and nutritional interventions, as well as varying lengths, doses, frequencies, and durations. This heterogeneity may have affected the pooled effect and the generalizability of the results. Secondly, this study limited the scope of the multidomain intervention components, aiming to reduce the heterogeneity by focusing on specific combinations. However, this approach inevitably excludes literature that includes other non-pharmacological interventions (such as psychosocial or sleep interventions), which may lead to a ‘sliced literature’ effect. Lastly, due to the paucity of available data, this study assessed only one physical function and compared the effects of multidomain interventions with single nutritional interventions. The limited number of studies available for this comparison may have compromised the robustness of the findings.

Implications for practice and research

Multidomain interventions are non-invasive, demonstrate good compliance, and can be recommended as a routine measure to enhance and maintain cognitive function in older adults. The moderate effect sizes observed for executive function and memory highlight the potential clinical significance of multidomain interventions. These findings support their integration into public health programs targeting cognitive decline prevention in older adults. Future research with rigorous designs should focus on comparing the effectiveness of multidomain interventions to single nutritional, exercise, and cognitive interventions. In addition, a more comprehensive assessment approach should be adopted to evaluate the effects of multidomain interventions. Moreover, more meta-analyses should include more articles to strengthen the statistical power and generalizability of the results.

Conclusions

The present study, involving six RCTs, investigated the effects of multidomain interventions on cognitive and physical functions. The results demonstrated that multidomain interventions significantly enhanced executive function and memory compared to the control group and nutritional interventions. This study further validates the effects of multidomain interventions on older adults with SCD and provides valuable insights for both clinical practice and future research. However, the impacts of multidomain interventions on global cognition and physical performance are inconsistent. Future research should focus on more robust multi-armed studies to explore the effects and underlying mechanisms of multidomain interventions.

Supplemental Information

Supplemental Information 1 Search Strategy.

Supplemental Information 2 Sensitivity analysis.

Supplemental Information 3 Publication bias of global cognition.

Supplemental Information 4 Publication bias of memory.

Supplemental Information 5 Publication bias of executive function.

Supplemental Information 6 Raw data.

Supplemental Information 7 PRISMA 2020 checklist.

Supplemental Information 8 The rationale for conducting the meta-analysis.

Additional Information and Declarations

Competing Interests

The authors declare that they have no competing interests.

Author Contributions

Qing Yi conceived and designed the experiments, performed the experiments, analyzed the data, prepared figures and/or tables, authored or reviewed drafts of the article, and approved the final draft.

Wei Wang conceived and designed the experiments, prepared figures and/or tables, authored or reviewed drafts of the article, and approved the final draft.

Yufei Qi performed the experiments, authored or reviewed drafts of the article, and approved the final draft.

Chengwei Yang performed the experiments, analyzed the data, prepared figures and/or tables, and approved the final draft.

Mengyun Sui analyzed the data, authored or reviewed drafts of the article, and approved the final draft.

Kun Meng conceived and designed the experiments, authored or reviewed drafts of the article, and approved the final draft.

Shanguang Zhao conceived and designed the experiments, authored or reviewed drafts of the article, and approved the final draft.

Data Availability

The following information was supplied regarding data availability:

The raw data is available in the Supplemental File. This is a sytematic review/meta-analysis.

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
