# Peer review of "The impact of multidomain interventions on cognitive and physical function in older adults with subjective cognitive decline: a meta‑analysis and systematic review"

_PeerJ, doi:10.7717/peerj.19588_

## Round 0.1 · original submission · Major Revisions

· Academic Editor

Major Revisions

Dear authors,

Reviewers have now commented on your paper. You will see that they are advising major revisions. If you are prepared to undertake the work required, I would be willing to consider a revised version.

·

Basic reporting

The submission must adhere to all PeerJ policies (see: 'Journal Policies'). Authors should provide the following information in a Supplemental File: 1. The rationale for conducting the meta-analysis is not provided.

Figures should be relevant to the content of the article, of sufficient resolution, and appropriately described and labeled. The figures provided are in a non-vectorial format (.png). This limits their usability as the reader zooms in to see the details. Plus, a brief description/explanation would be advisable to include as a foot note in both tables and figures.

The data on which the conclusions are based must be provided or made available in an acceptable discipline-specific repository. Raw data is not provided.

The present review aims to examine the effect of multidomain interventions on cognitive and physical function in people with subjective cognitive decline. The work entails a broad topic, tries to answer an important public health question and targets a population at a potentially strategic window for intervention. Although the field has been reviewed recently (Mohanty et al. 2023), the authors now add a quantitative meta-analysis of available literature, which a priori is a good reason for a review update. The authors have done a great work and merit congratulations. However, some methodological shortcomings need to be carefully revised in order to offer a clear preliminary quatitative answer to the matter. I offer the forthcoming comments for the editor and authors´ appraisal regarding the background, statistical analyses and result interpretation, hopefully helping to improve the manuscripts clarity and the validity of the findings.

- Some grammatical errors are present throughout the manuscript. A few examples are: line 214: “Table 1. Presents six eligible studies that involved 1.767…” or line 337: “the efficacy of multidomain interventions…” or line 352: “…the intervention period in this study is short..”. Please check for this kind of grammatical and spelling mistakes.

- In the backgraund section of the abstract the search of four databases is claimed, but actually five databases were searched.

The introduction adequately sets the subject of study and is well structured, overall. However, I suggest the folowing relevant modifications to ensure further clarity:

- The authors introduce dementia syndrome as a significant public health concern, and then appropiately define SCD. Yet, they switch to the statement in line 67 that says “SCD is regarded as the earliest precursor stage of Alzheimer´s disease…” and then end up stating that SCD offers an ideal window to delay the onset of dementia. Lifestyle strategies are promising interventions to prevent cognitive decline, and improve various aspects related to brain health. Whether multidomain interventions impact the core biological features of AD or delay the onset of dementia due to AD is still explicitly unproven, to my knowledge. Please, provide clarification in the text; are we talking about AD specifically, or about cognitive decline and dementia syndrome?

- Please, use updated literature to support the stated ideas in lines 62-65, 72-73 (this one exclusively refers to AD as well) and 74-75.

- Please, provide further bibliography to complete the prevalent view on the stated ideas (for example, in lines 76-77 of the manuscript, a sole reference published in 2012 –Morris, 2012– is supplied to support the effect of nutrition on brain health; this seems a bit too vague to me).

- It would be very nice if the authors expand upon the nuances in nutrition, exercise and cognitive training interventions for cognitive decline and sum up the main conclusions holding in each of those field of studies. Lines 77-86.

- As exergaming is not included in the present review, I would suggest to exclude it from the manuscript introduction. Lines 87-88.

- As the authors pertinently show, one of the biggest efforts to offer answers in the field of non-pharmacological interventions and dementia prevention stems from the original FINGER trial and the subsequently born World Wide FINGERS network. Recently, other FINGER-like trials have been completed and published with a variety of results and conclusions (i.e. the Agewell.de trial or the MIND-AD trial, among other). I would suggest the authors to comment on the most relevant findings of these studies, as they could provide further ground on the prevention of cognitive decline and serve as an adequate comparator for discussion. Lines 95-99

Experimental design

The review is clearly organized and structured, the research question is broad but explicit and the methodological sequence is adequately performed.

- I would argue that according to the multidomain intervention definition, all non-pharmacological interventions and intervention combinations should be gathered and analyzed systematically. The utilized inclusion criteria suggests a narrower, less comprehensive hypothesis, and thus a substantially different one, focusing on exercise + nutrition + cognitive training; and excluding other common interventions like the promotion of socioemotional health. Maybe defining multidomain as exercise + nutrition + cognitive triaining early in the manuscript and aims of the paper would stablish a clearer research context for the reader.

Validity of the findings

- Very few articles (k=6) with dissimilar outcome measures were included. An update of the search is recommended to capture recent available research (as the previous one was performed in March, 2024).

- The whole paper and reporting of the results are based on statistical significance and hypothesis testing. I strongly suggest that effect sizes, confidence intervals (or other measure of effect variablity) and the implication of the observed effects were included and discussed throughout the manuscript. How relevant could the delineated effect for policies and practice be?

- Excluding a bias domain due to its inherent presence within an intervention type is, I believe, unjustified. The authors should instead highlight it as a potential source of bias.

- Statistical Analysis: The use of a fixed-effects model assumes a single true effect size underlying all included studies. This assumption is particularly tenuous due to inherent heterogeneity stemming from the subjective nature of SCD and variability across sample populations and the disimmilar multidomain intervention approaches. Thus, a fixed-effects model may not be ideal here, as the single true effect assumption is unlikely to hold. Despite the common practice of fitting a fixed-effect model when I2 is less than 50%, the model selection should not be based on the test for heterogeneity (Borenstein, Introduction to meta-analysis). A nuanced approach that considers study-level characteristics rather than I2 statistic alone is essential and makes the use of random-effects models a better option in this case. Ideally, the Hartung-Knapp adjustment should be introduced, as it offers smaller type I error rates when very few studies are included. However, this would probably compute a very unprecise and uninforative effect size estimate (wider confidence intervals) and, if so, could be disregarded. Please consider restructuring analyses based on the random-effects model. A fixed-effects model could also be added as a sensitivity analysis, when appropriate (I2 <50%). I would also suggest the authors to then specify why results differ under each model and contextualize these differences based on study characteristics and quality in the discusión section.

- In the section “Effects of multidomain intervention versus control group”, precisely in the paragraphs describing the effect on global cognition and executive function, it is stated that the intervention produced a “significant” effect, while the effect is actually not statistically significant (i.e. p= 0.15 global cognition; p=0.06 executive function). The last one may be described as a trend. Please use significant when statistically significant in this context.

- Is there any identified exercise parameters recommended for the prevention of cognitive decline? Are included study interventions applying those ´dosages´? What about intensity? I believe discussing this kind of matter would deeply enhance the review.

- The discussion section of the manuscript goes through the findings, compare them to other relevant studies, engages in highlighting new research gaps and offers a precise overall conclusion.

Additional comments

no comment

Reviewer 2 ·

Basic reporting

What's the meaning of multidomain intervention in the study? please provide it.

The study seems to focus on nutrition in combination with other interventions, which is reflected in the inclusion of both nutritional and exercise intervention terms. Why do you not focus on Exercise intervention terms and mental or cognition exercise?

Therefore, the title should be the combination of nutrition with other intervention.

Experimental design

N/A
This is a systematic review and meta-analysis.
Line 155: why did you include both exercise and nutritional interventions with or without cognitive intervention? It seems that the main inclusion criteria were exercise and nutrition. The multidomain intervention might have exercise and cognition but not nutrition. Why did you focus on nutrition?

According to the search terms, the database was retrieved with #2 AND#3.

Validity of the findings

Table 1, please provide the findings in Table 1.
Discussion part: Regarding physical function, significance was not improved. You also explained that only one study was 30 min whereas other studies were 2 times/ week and intervention was not supervised. As you can see in Table 1, the intervention of exercise was 35 min with AE (Blumenth et al., 2020), supervision 3 times/week, and 60 min walking and strengthing exercise (Chatterjee et al., 2022), also 3 studies were resistance exercise with walking. Therefore, this might not be explained in the same way.
In addition, I suggest you add the parameter outcomes of the physical function. Regarding the exercise intervention, please rewrite with FITT (i.e., frequency, intensity, time, and mode of exercise). How about the intensity of exercise? Were those moderate intensity?

---

## Round 0.2 · Major Revisions

· Academic Editor

Major Revisions

Dear authors, there are still concerns regarding the readiness of your paper. Therefore, I request you to carefully address reviewer 1's comments.

·

Basic reporting

Basic reporting (journal policies and data access)

Adequately provided.

The included figures are not fully self-explanatory as the authors have mentioned. Indeed, it is not possible to directly identify if the comparator group is a ´control group´ or a nutritional intervention group (as is the case for figures 7 and 8). I would insist in providing a brief description in every figure.

Language and Style:

The authors have done a commendable job addressing the proposed modifications. However, significant grammatical and language issues persist throughout the manuscript, which impact the overall clarity and readability. At this stage of the review process, these issues should have already been resolved. Therefore, I would strongly recommend that the authors carefully review the entire text, ideally with the assistance of a colleague who is proficient in English or a professional language editing service. Improving the language will enhance the overall presentation of the manuscript.

To illustrate the type of issues that need attention, I provide a few specific examples:

• Line 45: “will be” – Since the inclusion of studies is already completed, this should be written in the past tense.
• Line 49: “…but did not significantly impact on global cognition…” – The preposition "on" is unnecessary here.
• Lines 53–55: “Further studies with robust designs…” – Subject-verb agreement and proper wording are needed.
• Line 95: “Some systematic reviews have proved the positive efficacy…” – Efficacy can not be positive or negative, either it exists (i.e., the intervention works) or it does not.
• Line 337: “the efficacy of multidomain interventions…” – Since multiple types of interventions are involved, "multidomain interventions" in plural is a more accurate and consistent term. The same applies to "cognitive interventions," etc.
• Lines 135-137 Written as it is, this sentence should go in the conclusion section of the manuscript.
• Similar issues appear in lines 102, 104, 105, 109, 117, 185, 187, 292-293, 300 & 327 (“significant, positive efficacy”, efficacy can not be positive or negative); 319-323 (could not be…); and 352 ("…the intervention period in this study is short…" should be adjusted for tense consistency); 354-356 & 360-362 are redundant; 397 “contributes…”.


The introduction has been adequately modified and is well structured, overall. However, I would suggest the following specific changes for the authors to consider:

- Dementia syndrome is appropriately introduced as a significant public health concern, and subjective cognitive decline is clearly defined. However, the statement on line 67 that “SCD is regarded as the earliest precursor stage of Alzheimer’s disease…” may not be entirely accurate, as some individuals show no subjective complaints at the time of mild cognitive impairment diagnosis. Thus, it may be an early clinical marker of Alzheimer´s and related dementias, but it is not universally present.

- Line 68. Then a meta analysis by Mitchell et al. 2014 is referenced to highlight the hightened risk of conversion to dementia holded by people with SCD or MCI. It would be nice if the authors offered a brief description of the core meta-analytical design and relative risk results of this work (i.e. observational cohorts, risk comparisons related to those those considered to have a healthy cognitive aging…).

- Line 70. The use of the term “ideal” is a bit controversial, I believe. “Promising” would arguably be more accurate in this case.

- Line 75. A meta-analysis including subjects with MCI is referenced (Teixeira et al. 2012). Importantly, the effect of non-pharmacological interventions may differ between cognitively healthy older adults, people with SCD and those with MCI. For example, current evidence suggest that the effects of exercise may vary between people with and without MCI (Falck et al. 2019). If no better suitable reference is available, the difference above should be clearly reflected in the text. In addition, I would suggest to add the meta-analysis performed by Hafdi et al. 2021 regarding the effect of multidomain interentions on cognitive outcomes, published by the Cochrane Collaboration, as an important reference. Our group also showed differences between long-term exercise interventions and multidomain interventions on cognitive function, in a recently published article.

- Lines 108-111. The reference included by the authors in the revised version of the manuscript is a secondary analysis that showed that the risk of dementia (indirectly measured via a dementia-risk index) was effectively diminished in the intervention group. Nevertheless, the primary outcome (change in global cognition composite Z) of the AgeWell.de trial was not met. Of course, this needs to be clearly reflected in the text to show that inconsistencies remain in the field of non-pharmacological multidomain dementia prevention strategies. Furthermore, the J-MINT trial could also be commented upon, due to it being a published finger network trial.

Experimental design

Experimental Design

The review is clearly organized and structured, and the methodological sequence is adequately performed.

- I would still argue that according to the multidomain intervention definition, all non-pharmacological interventions and intervention combinations should be gathered and analyzed systematically. Defining multidomain as exercise + nutrition + cognitive triaining early in the manuscript, as the authors have now adequately delineated, provides a clear research context for the reader. Nevertheless, to adress such specific questions by the means of a systematic review and meta-analytical approach often produces what has been called a “sliced literature”, as critically signalled by other authors in previous works (see Ciria et al 2021 for a well-conducted comprehensive work and brief critique of the consequences of this problem). I suggest this could be discussed as a limitation in the limitations section.

- Line 168. If the previous available systematic review included people older than 55 years (Kumar 2022), why did the authors select a different threshold for age (45 years)? Please, provide an explanation for this decision.

- Lines 184-188. The first sentence needs to be written in past tense. Then, “Secondary outcomes: the physical function including the physical performance” sentence is largely confusing for the reader, what´s the difference?

- Line 148. If the search was effectively updated, please update the information related to date of search accordingly.

- Line 218. The symbol (>) is not the correct one, please modify it.

- Lines 223-225. After removing 1.080 duplicates 4.322 articles would remain for screening, not 4.189 as stated, correct?

- Line 300-301. p=0.06… for the observed significant effect on executive function. The oversight persists.

- Line 312-313. The explicit difference between the physical function and physical performance concepts is not presented in the manuscript.

- Statistical Analysis: I am glad to see that the authors have considered the recommended statistical approach based on a random effects model and have conducted the needed analysis. Their findings, although preliminary, are now far more robust.

Validity of the findings

The discussion section of the manuscript effectively presents the findings, compares them with other relevant studies, identifies new research gaps, and provides a reasonably precise overall conclusion. However, I strongly recommend that any statements regarding mechanistic evidence in humans or animal models be explicitly discussed within the context of the specific model used (e.g., "Kempermann et al., 1997 showed that an enriched environment helps preserve the integrity of white matter connections – in mouse models). This would help avoid ambiguity and improve clarity.

Reviewer 2 ·

Basic reporting

The revised manuscript is clear and could be accepted for publication.

Experimental design

N/A

Validity of the findings

The revised manuscript is clear and could be accepted for publication.

---

## Round 0.3 · accepted · Accept

· Academic Editor

Accept

The reviewers have agreed that the work made by the authors is overall suitable for publication.

·

Basic reporting

No comment

Experimental design

No comment

Validity of the findings

No comment

Additional comments

Mild readability issues and little grammatical mistakes persist. Otherwise, the work made by the authors is overall suitable for publication.